# Sorption of Bisphenol A as Model for Sorption Ability of Organoclays

**DOI:** 10.3390/molecules27144343

**Published:** 2022-07-06

**Authors:** Issaka Garikoé, Boubié Guel, Ingmar Persson

**Affiliations:** 1Laboratory of Materials and Molecular Chemistry, U.F.R–SEA, University Joseph KI-ZERBO, Ouagadougou 03 BP 7021, Burkina Faso; garikoe@yahoo.fr; 2Department of Molecular Sciences, Swedish University of Agricultural Sciences, P.O. Box 7015, SE-750 07 Uppsala, Sweden; ingmar.persson@slu.se

**Keywords:** smectite, organoclays, surfactant, intercalation, hydrophobicity, rearrangement, expansion, interaction, partitioning process, Ca-montmorillonite

## Abstract

The arrangement of bisphenol A molecules into organoclays and their interactions with the intercalated surfactant were studied. The organoclays were prepared via solid-state intercalation of four cationic surfactants, such as dodecyltrimethyl-, tetradecyltrimethyl-, hexadecyltrimethyl-, and didodecyldimethyl-ammonium, as bromide salts, at different loading levels into the interlayers of two natural clays. The natural clays, the prepared organoclays, and the spent organoclays were characterized by X-ray powder diffraction, infrared spectroscopy, and scanning electron microscopy. X-ray powder diffraction measurements showed successive interlayer expansions of the *d*_001_ basal spacing due to the intercalation of the cationic surfactants and the bisphenol A sorption. The increased *d*_001_ basal spacing of the organoclays after bisphenol A sorption indicates that the molecules are integrated between the alkyl chains of the surfactant in the organoclays interlayers. Infrared spectroscopy was employed to probe the intercalation of the cationic surfactants and the sorbed bisphenol A. New characteristic bands attributed to the bisphenol A phenol rings appear in the range 1518–1613 cm^−1^ on the infrared spectra of the spent organoclays, proving the presence of bisphenol A in the hydrophobic interlayers. Scanning electron microscopy of the organoclays before and after BPA sorption shows that their morphology becomes fluffy and that the presence of the organic molecules expands the clay structure.

## 1. Introduction

Bisphenol A (BPA) is one of the endocrine disruption compounds (EDCs) that can cause a hazard to human health at a low concentrations [1,2]. Large volumes of BPA are used in the industrial production of epoxy resins and polycarbonates [1,2,3]. This has led to high demands for BPA in the production of polycarbonates and thereby linked to human health [2,4]. BPA is reported to affect estrogenic activities and induces liver damage, pancreatic β-cell function disruption, thyroid hormone disruption, obesity-promoting, diabetes, etc. [4]. This is related to continuous exposure to BPA through food, drinking water, dental sealants, dermal exposure, and inhalation. The octanol-water partition coefficient (*K*_ow_) of BPA was determined to be around 3.32. As a result, BPA shows strong hydrophobicity and, thereby, low water solubility (120–300 mg/L) at 25 °C. BPA was found to be one of the emerging pollutants widely detected in the environment, and the migration of BPA to the aquatic environment is a major concern. Thus, due to the negative effects of BPA on human health mentioned above, many studies were conducted to investigate the removal of BPA from aqueous sources by the implementation of appropriate, reliable, and safe water treatment processes.

From different treatment techniques, such as chemical advanced oxidation [5], membrane filtration [6], and electrochemical mineralization [7], sorption is strongly favored because of its simplicity, cost-effectiveness, ease of operation, and high efficiency. Different developed methodologies are still improving the sorption efficiency of organically modified clays in terms of achieving high sorption efficiency to remove organic pollutants such as BPA from water. Removal of BPA from aqueous solutions by sorption using organoclays synthesized from natural smectite containing clays with different types of organic cationic surfactants was demonstrated in several investigations [8,9,10,11,12,13]. A review of scientific studies on the BPA removal from water by sorption methodology was published recently [14]. All studies reported a high ability of organoclays to sorb BPA compared to the untreated clays [8,9,10,11,12,13,14]. The organoclays are described as efficient sorbents expressed in terms of sorption capacity [8,11,13,15,16]. In our recent paper, we have shown that organoclays prepared by using two natural smectites clays (denoted AH and DI) and four cationic surfactants were effective sorbents for BPA removal from water [11].

From the literature, various parameters, such as the effects of sorbent dose, agitation time and kinetics, initial concentration and isotherm, pH, surfactant loading levels, temperature, and thermodynamics, were considered in the study of BPA sorption processes [8,11,16]. It has been shown that organoclays are efficient sorbents expressed in terms of sorption capacity, fitted isotherms, kinetic models, and thermodynamic aspects [8,11,13,16]. Previous studies reported that the sorption mechanism of BPA on organoclays is best described by pseudo-second-order kinetics and the Langmuir isotherm model, with maximum sorption capacity in acidic solution [8,11,13,16,17]. In our previous study we profoundly investigated the thermodynamic and kinetic parameters and found that physisorption and chemisorption might be involved in the BPA adsorption mechanism [11]. Our results suggested that organoclays prepared via solid-state intercalation could be considered as effective adsorbents as those obtained via solid-liquid reactions, for the removal of BPA molecules from water [11]. It has been shown that pH of the reaction mixture is the most important factor to determine the sorption properties of the sorbent, and acidic conditions seem ideal for the sorption process. The behavior of BPA depending on pH values is presented in Appendix A [18]. The negative surface charge of the clays and the deprotonation of BPA in an aqueous solution with increasing pH are certainly important factors in the decrease in sorption capacity when the pH reaches 10 [9,17]. Thus, the decrease in the sorption capacity of the modified clays seen in the alkaline pH range can be explained by the electrostatic repulsion between the negative charge of the clays surface and the bisphenolate anions [17], and stronger hydration of the phenolate groups than the phenol ones makes it difficult for BPA to enter the hydrophobic interlayers. Similar results have been reported by Park et al. [8]; Garikoé et al. [11]; Dong et al. [17]; and Xu et al. [9]. Although many studies have been devoted to the efficiency of synthesized organoclays on BPA sorption at the batch scale [8,11,13,16], few investigations are available regarding the quantification of interactions between mineral phases, surfactants, and BPA molecules present into the organoclays interlayers. Consequently, more studies are needed in terms of characterization techniques such as X-ray powder diffraction (XRPD), Fourier transform infrared (FT-IR) spectroscopy, and scanning electron microscopy (SEM), to analyze the sorption process of BPA into the organoclays interlayers. This includes BPA molecules arrangement in the interlayers, their interactions with the intercalated surfactants and subsequently the rearrangement of the surfactant chains within the interlayers after secondary BPA sorption. It is worth highlighting the fact that XRPD analysis, infrared spectroscopy and SEM analysis are among the most informative techniques used to investigate the adsorption of BPA on organoclays and to obtain information on the molecular arrangement in their interlayers and finally to examine the rearrangements of the surfactants after the second BPA adsorption. The present paper constitutes a follow-up of our previous paper [11] and aims to provide detailed experimental results for a better understanding of the interfaces BPA sorbate/cationic surfactants/Ca-montmorillonite.

In the literature, FT-IR spectroscopy was used to prove the presence of certain functional groups on the surface of spent graphene after BPA sorption [9]. It was reported that the peak related to the characteristic vibration of the aromatic C=C bonds appeared at around 1622 cm^−1^ in the infrared spectra of the spent graphene, sustaining the BPA sorption onto the adsorbent [9].

XRPD analysis was applied to determine the basal spacing of organoclays after sorption of chlorophenols (4-chlorophenol and 2,4-dichlorophenol) into *n*-C_16_H_33_(CH_3_)_3_N^+^ and *n*-C_12_H_25_(CH_3_)_3_N^+^ surfactant modified montmorillonite [19]. A slight decrease of the *d*_001_ basal spacing of organoclays after 4-chlorophenol and 2,4-dichlorophenol sorption was reported [19]. The decrease of the *d*_001_ basal spacing after chlorophenol sorption was attributed to the formation of tightly packed aggregates with van der Waals and electrostatic forces [19]. Ghavami et al. [20] reported an interlayer expansion of C_16_-montmorillonite after naphthalene sorption, which indicated that naphthalene molecules penetrated and remained in the C_16_-montmorillonite interlayers [20]. Similarly, they found that the *d*_001_ basal spacing of organo-bentonites increased upon the sorption of C_6_ and C_8_ alkanes and that the sorbate molecules penetrated the interlayer [20]. A similar observation was made for the 2-chloroaniline sorption into organically modified clays [21]. A previous study indicated that the sorption of alachlor and metolachlor herbicides caused an increase in the interlayer space of sebacate anion modified layered double hydroxide (LDH), but on the contrary, their sorption onto dodecylsulfate anion modified layered double hydroxide did not cause any significant change in the interlayer spacing [22]. Liu et al. reported a marginal increase in the *d*_001_ basal spacing of the organoclays after BPA sorption and concluded that BPA molecules were not intercalated into the organoclays interlayers [23]. According to these results, it can be noted that the impact of the organic sorbates on the organoclays interlayer, as indicated by the variation of the *d*_001_ basal spacing, seems to be linked to the type of organoclays, the chemical structure of the surfactant being intercalated and the type of the sorbed organic sorbates.

SEM analysis was used to investigate the morphological differences in the SEM images before and after BPA sorption [22,24]. It has been claimed that surfactant molecules can also be adsorbed on the outer surface of the clays, and this occurs on the hydrophobic surface of the organoclays, which provides suitable sites for BPA sorption [13].

It is then clear that XRPD, FT-IR, and SEM analyses are valuable techniques to probe the impact of organic sorbates on surfactants-modified organoclays. The structural geometry and texture of organoclays are probed using XRPD analysis. The molecular arrangement of the intercalated surfactants and BPA molecules are well understood by FT-IR investigations. The morphological changes of the clay composites are assessed by SEM pictures. However, there are very few studies, if any at all to our knowledge, using all of these techniques together to provide in-depth information regarding BPA arrangements and interactions with surfactants in the organoclays interlayers.

This study aims to get a better understanding of (i) the sorbed BPA molecules’ arrangement in the organoclay’s interlayers; (ii) the interactions of the sorbed molecules with the surfactants in the interlayers of the organoclays; and (iii) the surfactant chains rearrangement after BPA sorption. A detailed study, using laboratory characterization (XRPD, FT-IR, and SEM), was carried out on spent organoclays to investigate the change of the organoclay’s interlayers microstructure.

## 2. Materials and Experimental Methods

### 2.1. Materials

The surfactants *n*-dodecyltrimethylammonium bromide, *n*-C_12_H_25_(CH_3_)_3_NBr, (C_12_), *n*-tetradecyltrimethylammonium bromide, *n*-C_14_H_29_(CH_3_)_3_NBr, (C_14_), *n*-hexadecyltrimethyl-ammonium bromide, *n*-C_16_H_33_(CH_3_)_3_NBr, (C_16_), di-*n*-dodecyldimethylammonium bromide, (*n*-C_12_H_25_)_2_(CH_3_)_2_NBr, (2C_12_), and bisphenol A (C_15_H_16_O_2_, ≥99% purity) were purchased from Sigma-Aldrich and used without further purification. The natural clays used in this study are from Siétougou and Diabari villages, denoted AH and DI, respectively, located in the Eastern region of Burkina Faso at the following coordinates, AH: latitude = 11°54′43.2″, longitude = 00°38′22.7″, and DI: latitude = 11°51′19.9″, longitude = 00°25′08.07″.

### 2.2. Experimental Methods

Organoclays preparation: The organoclays have been prepared according to the previously reported methodology [25] and denoted as surfactant loading-type of the surfactant-original location of the clay, e.g., 1.0 CEC-C_12_-AH. Detailed procedure for obtaining organoclays are given in the SI section. The experimental procedure on BPA adsorption using the raw clays and their organically modified clays is also detailed in the SI section.

Spent organoclays collection: After BPA sorption, the spent clays and spent organoclays were collected, dried at room temperature, and further dried in an oven at 105 °C for 24 h. The dried spent clays are labeled as BPA-original location of clay and the spent organoclays are labeled as surfactant loading-type of the surfactant-BPA-original location of clay, e.g., BPA-AH and 1.0 CEC-C_12_-BPA-AH, respectively.

Characterizations of the spent organoclays: The BPA spent clays (or organoclays) were characterized by XRPD, FT-IR, and SEM. The XRPD patterns of the samples were recorded in the angle range 1.06–32° with a step size of 0.2° in scattering angle, 2*θ*, by a *θ-θ* goniometer as described elsewhere [26] using Mo(Kα1) radiation, λ = 0.71073 Å. The raw clays, the organoclays, the spent clays (or organoclays), the pure surfactants, and the pure BPA were mixed with dried KBr and pressed into discs with 2 mg of sample and 200 mg of KBr in each tablet for the FT-IR measurements. The data were recorded on a Perkin Elmer Spectrum 100 FT-IR spectrometer over the spectral range of 400–4000 cm^−1^. The SEM analyses were performed on a Hitachi S-3400 N instrument, with a resolution of 25.0 nm in the variable pressure mode.

## 3. Results and Discussion

### 3.1. Characterization of BPA-Clays and BPA-Organoclays

#### 3.1.1. X-ray Powder Diffraction

XRPD diffractograms of the natural raw clays and the organoclays before and after BPA sorption are shown in Figure 1, Figure 2, Figure 3, Figure 4 and Figure 5. The diffractograms of the AH and DI raw clays (Figure 1) show that they contain montmorillonite, kaolinite, quartz, orthoclase, hematite, and rutile, as well as anorthite in the DI clay [25]. Figure 1 shows that there is only a marginal increase in the *d*_001_ values of the Ca-montmorillonite of the natural raw clays (AH, DI) after treatment with an aqueous solution of BPA. The slight increase in *d*_001_ basal spacing, from 15.3 to 16.5 Å shows that only a few BPA molecules penetrated into the interlayer space of the natural raw clays, and may suggest sorption on the external surfaces of the clay, probably at the broken edges [27]. The slight increase of the *d*_001_ values sustained less adsorption efficiency of the raw clays towards BPA and this result is in agreement with the results obtained in our previous study by using a High-Performance Liquid Chromatography dosage of BPA [11].

It is worth mentioning the fact that in addition to montmorillonite, AH and DI clays contain non-swelling phases, such as kaolinite, orthoclase, and anorthite, as well as other minerals phases, such as quartz, rutile, and hematite. However, literature results indicated that all these phases contributed weakly in the removal of organic pollutants. For example, Styszko et al. and Alkaram et al. reported that unmodified montmorillonite and bentonite or kaolinite had a low adsorption capacity toward phenolic compounds [28,29]. According to the work by Li et al., iron oxides such as hematite (α-Fe_2_O_3_), maghemite (γ-Fe_2_O_3_), and lepidocrocite (γ-FeOOH) presented a low sorption capacity toward BPA (2.7%) in absence of UV irradiation [30]. However, UV irradiation caused a photo-degradation of the BPA molecules. Kaplan et al. [31] reported a low percentage of BPA sorption (5%) by rutile from aqueous solution of 10 mg/L of BPA as initial concentration. The authors brought back an improvement of the rate of mineralization of the BPA (51%) in the presence of UV irradiation. Taking into consideration all these above results, it is then concluded that in absence of UV irradiation, hematite and rutile, which are present in our AH and DI clays, would contribute very little to the adsorption of the BPA. As a result, the removal of BPA is mainly due to its interaction with the surfactant modified organoclays, as reported [11].

The diffractograms of the spent organoclays, seen in Figure 2, Figure 3, Figure 4 and Figure 5, indicate that it is only the 001 reflection of the Ca-montmorillonite which shifts to lower 2*θ* angle values. Appendix A summarized the *d*_001_ values of organoclays before and after BPA sorption.

The XRPD diffractograms, given in Figure 2, Figure 3, Figure 4 and Figure 5, indicate that the *d*_001_ basal spacing of the organoclays increases significantly after BPA sorption. Such an increase in the *d*_001_ values sustains the high sorption efficiency of the organoclays towards BPA molecules and this result is in agreement with our previous work [11]. This increase also proves that the BPA molecules are interacting with the alkyl chains of the surfactants in such a way that the interlayer expands, as proposed elsewhere [25,32]. Contrary to previous literature results reporting that only a small increase in the basal spacing was observed after sorption of BPA, which lead to the conclusion that BPA molecules were not intercalated into the interlayers of the organoclays [23]. However, the present study indicates that the *d*_001_ basal spacing of the organoclays increases significantly after sorption of BPA, and shows that BPA molecules enter the interlayers of the investigated organoclays. This increase in the *d*_001_ basal spacing is similar to the results obtained by Ghavami et al. on naphthalene and petroleum hydrocarbons sorption on C_16_-montmorillonite and organo-bentonites [20] and 2-chloroaniline sorption on dimethyl ditallowylammonium modified clays [21]. The insertion of the BPA molecules into the organoclays interlayers causes a rearrangement of the intercalated surfactants which raise in the same way as found at increased surfactant loading [25]. The surfactant molecule orientation in the interlayer is strongly dependent on the degree of loading with the alkyl chains oriented parallel with the sheets at low loading, but the long alkyl chains are rising with increasing loading to be standing almost perpendicular to the sheets at high loading levels [25,32]. Ghavami et al. reported an increase of 2 Å in the *d*_001_ basal spacing of C_16_-montmorillonite after naphthalene sorption and this indicated that naphthalene molecules penetrated and remained in the C_16_-montmorillonite interlayers [20]. This result supports the fact that naphthalene sorption occurs mostly in the organoclays interlayer and the authors assumed that the sorption is driven by the short-range forces between sorbates and surfactant sorbents [20]. The crystal structures of the surfactants (*n*-C_12_H_25_(CH_3_)_3_N)Br and (*n*-C_12_H_25_(CH_3_)_3_N)Br) with incorporated *p*-phenylphenol have been reported [33]. According to this study, these crystal structures showed that the *p*-phenylphenol molecules are sandwiched between the dodecyl (*n*-C_12_H_25_) chains. It seems appropriate to draw an analogy with this study and suggest that the BPA molecules are arranged similarly in the interlayers of the organoclays which are investigated in the present work. The increase of the basal spacing also indicates that physisorption is the major process involved in the sorption reaction rather than chemisorption.

In our previous study [11], we reported that the amount of BPA sorbed on the investigated organoclays remained approximatively constant when 2 ≤ pH ≤ 8 (*q_e_* values ranged from 19.8 to 19.4 mg/g), decreased weakly when 8 ≤ pH ≤ 10 (*q_e_* values ranged from 19.6 to 16.6 mg/g), and decreased remarkably when pH > 10 (*q_e_* values ranged from 18.5 to 9.0 mg/g). The BPA molecules remain neutral as long as pH ≤ 8 and interacts with alkyl chains of the surfactants through van der Waals forces. However, at pH > 8 BPA starts to deprotonate and can interact with the alkylammonium ions of the surfactant located in the interlayer of the organosmectites. The possibility for a charged species to enter a principal neutral organic interlayer is generally hampered. The ion-dipole or coordination-type bonding is possible with a hydrophobic surface but would be greatly retarded by the strong competition from water in the case of raw clay having a hydrophilic surface. In this case, the BPA molecules act as an acceptor of electrons donated from sites of negative charge in the mineral clay surfaces [33]. A possible interaction mechanism is that hydrogens of the phenol groups in BPA bind to Si-O or Al-O groups in clay mineral surfaces. However, the absence of significant sorption of BPA by the raw clays indicates that such mechanism is of only minor importance if any at all. One would expect a largely reduced sorption of BPA at alkaline pH provided that hydrophobic interaction is the sole mechanism. However, the affinity of organoclays for anionic species is not changing significantly when increasing pH from 8 to 10, which shows that there is an additional chemical interaction between the surfactant and anionic BPA. As the BPA molecules become charged starting at pH 8, an additional electrostatic interaction seems to take place between the charged BPA and the tetraalkylammonium group of the surfactant molecules. Such interaction will further strengthen the interaction between the surfactants and the charged BPA molecules. Figure 6 shows surfactant molecules and BPA molecules arrangements/orientations into the interlayer space of the Ca-montmorillonite. The proposed arrangement is based on previous models by Dong et al. [17] on BPA sorption onto surfactant modified zeolites and Kamitori et al. [33] on the C_12_ alkyl chains and *p*-phenylphenol complex study. The increase of the *d*_001_ basal spacing sustains that the BPA molecules enter into the organoclays interlayers and Figure 6 represents the possible arrangement/orientation of the BPA neutral or mono-ionized molecules into the interlayers. The BPA molecules are sandwiched between the long alkyl chains (C_12_, C_14_, C_16,_ and 2C_12_) of the surfactants and interact with them.

The BPA neutral molecules interact with the nitrogen atoms (positively charged) of the cations of alkylammonium by the intermediary of oxygens of the phenolic groups which may act as electron donors.

It can be seen in the proposed arrangement that for the neutral forms of BPA (pH ≤ 8), the hydroxyl groups (-OH) of the phenol ring in the BPA molecules tend to interact with the positively charged nitrogen of the surfactant (Figure 6a). In the case of the BPA mono-ionized forms (pH > 8), the -O^−^ groups of BPA tend to join the positively charged nitrogen of the surfactant, and the hydroxyl group (-OH) pointed inside of the hydrophobic phase created by the long alkyl chains (Figure 6b). The proposed orientation is due to the difficulty for the hydrated -O^−^ to enter the hydrophobic region of the long alkyl chains (Figure 6b).

It is important to remember that the investigated AH and DI clays contain, in addition to montmorillonite, non-swelling phases, such as kaolinite, orthoclase, anorthite, and other minerals phases (quartz, rutile, and hematite). However, these phases are only likely to contribute weakly to surface retention of the surfactant molecules, and consequently, they would not predominantly take part in the BPA adsorption onto their surface [28,34,35,36,37].

#### 3.1.2. Infrared Spectroscopy

The infrared spectra of pure BPA and natural raw clays before and after sorption of BPA are shown in Figure 7 and those of pure BPA, and organoclays before and after BPA sorption are shown in Figure 8 and Appendix A.

All spectra show that the Si-O-Si stretching vibration bands at around 1050 and 1100 cm^−1^ were not affected by BPA sorption. The bands at 918, 538, and 470 cm^−1^ assigned to the bending vibrations of Al-Al-OH, Si-O-Al, and Si-O-Si groups, respectively, were neither affected by BPA sorption [38]. Moreover, BPA molecules could be identified on the sorbents due to the observation of its characteristic bands on the infrared spectra. The broadband at 3350 cm^−1^ in the spectrum of pure BPA is assigned to O-H stretching self-H-bonded hydroxyl groups. According to the studies by Fei et al. [39] and Ullah et al. [40] on the spectrum of the pure BPA, there should be a dynamic equilibrium between the free hydroxyl (OH) groups and those having established hydrogen bonds [39,40]. The absence of OH bands above 3600 cm^−1^ in the infrared spectrum of pure BPA confirms that the hydroxyl groups form hydrogen bonds in pure BPA [9,39,40]. The doublet at 3077 and 3038 cm^−1^, seen on the pure BPA spectrum, is due to the stretching modes of the aromatic CH groups. The characteristic asymmetric and symmetric stretching vibrations of the CH_3_ group appear at 2929 and 2871 cm^−1^, respectively, and the *para*-substituted aromatic system gives well-defined bands at 1613, 1599, 1518, 1179, and 824 cm^−1^ in the spectrum of the pure BPA. The structure of the isopropyl unit in BPA is associated with the band at 1363 cm^−1^ [39,40]. The FT-IR spectra of the natural raw clays after BPA adsorption (Figure 7) show new peaks at 2857 and 2926 cm^−1^, which were assigned to the asymmetric and symmetric stretching vibrations of the methylene groups in the BPA molecule.

The intensities of the corresponding peaks after BPA adsorption on natural raw clays are weak, which indicates that the raw clays sorb only a small amounts of BPA as described elsewhere [11].

The FT-IR spectra of BPA-treated organoclays, Figure 8 and Appendix A, show new peaks proving the presence of BPA in the organoclays. The bands at 1626 and 1613 cm^−1^, only observed on the spectra of BPA-organoclays and the pure BPA, are allotted to vibrations of aromatic C=C bonds of the phenyl rings [9,38]. The bands around 1518 cm^−1^ are assigned to the aromatic CH vibrations. A broadening of the peak around 3350 cm^−1^ in the spectra of BPA-treated clays and organoclays towards lower wavenumber values is observed, which could be explained by the contribution of aromatic hydroxyls of the BPA engaged in hydrogen bonds. The peaks of AlMg-OH and Al_2_-OH seen at 3694 and 3620 cm^−1^, respectively, remain on the FT-IR spectra of the organoclays after BPA sorption [41,42,43]. These bands are normally assigned to stretching vibrations of hydroxyl groups coordinated to octahedral cations in kaolinite or montmorillonite. The wavenumbers of symmetric and asymmetric stretching vibration of CH_2_ groups of surfactants before and after BPA adsorption are summarized in Table 1.

The characteristic bands observed between 3670 and 3700 cm^−1^ in the spectra of the organosmectites are always present in the spectra of BPA-organoclays, indicating that, after sorption, the hydrophobic character of the organoclays is maintained. The band at 3620 cm^−1^ is not affected after the BPA molecules sorption. This indicates that the binding energy of OH is not reduced compared to the Al-Al-OH bond [38]. The appearance of the new bands on the spectra of BPA treated raw and organoclays clearly show the sorption of BPA.

It is seen that the wavenumbers generally shift weakly to high values (Table 1), and this is assigned to the rearrangement of the alkyl chains of the surfactant molecules (C_12_, C_14_, C_16_, and 2C_12_). This rearrangement of the molecules of surfactant can be explained by the interactions between the cationic surfactants and the BPA molecules in the interlayer as discussed above.

The interaction mechanism is described as a partitioning process into the organic solvent-like hydrophobic region created by the long alkyl chains of the surfactants. These bonds are of van der Waals type and are deemed to operate analogously in the sorption of BPA by organoclays. It is further suggested that, when interacting with each surfactant layer, the two hydrophobic phenyl rings of BPA molecules would point to the inside of the hydrophobic phase of surfactants, because this orientation allows the hydrophobic interaction between the phenyl rings of BPA and the C_12_, C_14_, C_16_, and 2C_12_ tail of the surfactants to come close to each other and thus strengthen the retention of BPA molecules.

#### 3.1.3. Scanning Electron Microscopy and Energy Dispersive Analysis of X-ray

Scanning electron microscopy (SEM) is a useful technique to assess the morphological changes of clay composites. The SEM coupled with an EDAX detector was used to evaluate the surface morphology and composition of the organoclays before and after BPA adsorption. SEM images and EDAX spectra of AH raw clay and the 1.0 CEC-C_12_-AH organoclays before and after BPA sorption are shown in Figure 9, Figure 10 and Figure 11, respectively.

Figure 9a, Figure 10a, and Figure 11a show that there are noticeable differences between the surface morphology of raw natural clays (Figure 9a) and organoclays (Figure 10a), whereas no such morphological differences were observed between organoclays before (Figure 10a) and after BPA sorption (Figure 11a). The AH raw clay SEM micrographs (Figure 9a) show a massive, curved plates and aggregated morphology [44,45]. However, the presence of the surfactants and the sorbed BPA expands the clay structure, which induces a more open morphology, Figure 10a and Figure 11a [44,45]. The organoclays are homogeneously fluffy [24] and their morphology remained after BPA sorption. Shattar et al. [46] reported that montmorillonite is a clay mineral phase with face-to-edges contacted between the particles, with various orientations, and without domains or clusters along the surface. These observations are in agreement with the XRPD results showing the sorption of BPA into organoclays. Simultaneously to SEM images, Energy Dispersive Analysis of X-ray spectroscopy (EDAX) was also performed. Figure 9b, Figure 10b, and Figure 11b revealed the presence of C, O, Al, Si, and Fe in the raw clay and the organoclays before and after BPA adsorption. The high percentage of carbon observed on the EDAX spectra of organoclays before (76.84%) and after BPA sorption (79.13%) compared to the raw clay (38.69%) may be attributed to the presence of cationic surfactants and BPA molecules in the organoclays interlayers.

### 3.2. Comparison of the Results with Those Obtained from Similar Previously Conducted Studies

A few studies have proposed arrangements of BPA molecules in the inner/outer layers of organoclays [8,23] or organozeolite [17]. BPA molecules retention mechanism and their interactions in the organoclays interlayers are interpreted in different ways in the literature [8,17,23]. Table 2 summarizes the comparison of the results in the present work with those obtained from similar previously conducted studies. Whereas the previous reported investigations indicated that BPA removal from aqueous solution occurred through inner and outer retention mechanism along with a marginal change in the *d*_001_ basal spacing, the present work showed a significant increase in the *d*_001_ basal spacing in the montmorillonite phase, which suggested that the adsorption of BPA molecules is mainly due to their insertion into the montmorillonite interlayers. In this context, it can be suggested that in the BPA removal the inner retention is predominantly favored compared to the outer retention.

## 4. Conclusions

This work is part of a global research on the sorption of bisphenol A by surfactants modified natural smectites clays minerals in order to purify waste-water. The sorption of BPA on the synthesized organoclays was carefully investigated by carrying out laboratory experiments such as XRPD, FT-IR and SEM analyses.

Significant results were obtained regarding XRPD analysis. It was shown that the *d*_001_ basal spacing of the organoclays increased significantly after BPA sorption and this strongly indicated that the BPA molecules penetrated the organoclays interlayers. The BPA neutral molecules and their mono-ionized forms are mainly inserted into the surfactants modified Ca-montmorillonite. A small part of BPA neutral molecules and their mono-ionized forms are retained onto the Ca-montmorillonite, kaolinite, and other impurities outer surface. These retentions are connected to the interaction of BPA molecular or their mono-ionized forms with the positively charged nitrogen of C_12_, C_14_, C_16_, and 2C_12_ surfactants on both the inner and outer layers.

The FT-IR analyses showed that the characteristic peaks related to the CH_2_ asymmetric and symmetric stretching vibration, seen at around 2920 and 2850 cm^−1^ on the organoclays spectra, are maintained after BPA sorption. It was also shown that the wavenumbers of the CH_2_ asymmetric and symmetric stretching vibrations shifted weakly to high values. This was assigned to the entering of the BPA molecules into the organoclays interlayers, the interactions between the BPA and the surfactant molecules, and the rearrangement of the surfactant long alkyl chains (*n*-C_12_, *n*-C_14_, n-C_16_, and *2n*-C_12_).

SEM images showed that the organoclays before and after BPA adsorption were homogeneously fluffy. EDAX spectra showed an increase in the carbon content upon the surfactant intercalation and the BPA sorption.

To fully assess the practical implications of the results described in this paper, the fate of the adsorbed BPA contaminant needs to be further investigated. Current investigations are being carried out in order to assess the possible re-use of the spent organoclays.

## Figures and Tables

**Figure 1 molecules-27-04343-f001:**
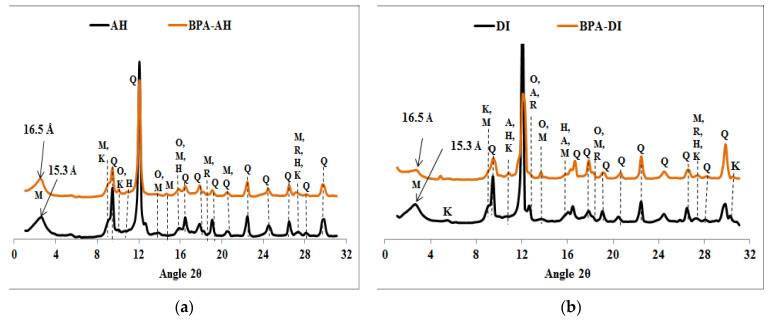
XRPD patterns of clays samples before and after BPA adsorption: (**a**) AH, (**b**) DI. Q = Quartz, M = Montmorillonite, K = Kaolinite, H = Hematite, R = Rutile, O = Orthoclase and A = Anorthite.

**Figure 2 molecules-27-04343-f002:**
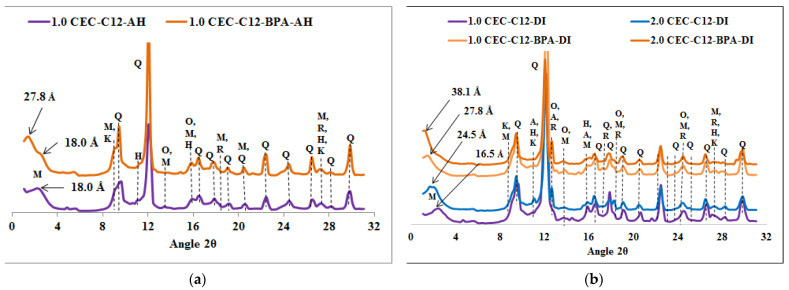
XRPD patterns of organoclays samples before and after BPA adsorption: (**a**) C_12_-AH, (**b**) C_12_-DI. Q = Quartz, M = Montmorillonite, K = Kaolinite, H = Hematite, R = Rutile, O = Orthoclase and A = Anorthite.

**Figure 3 molecules-27-04343-f003:**
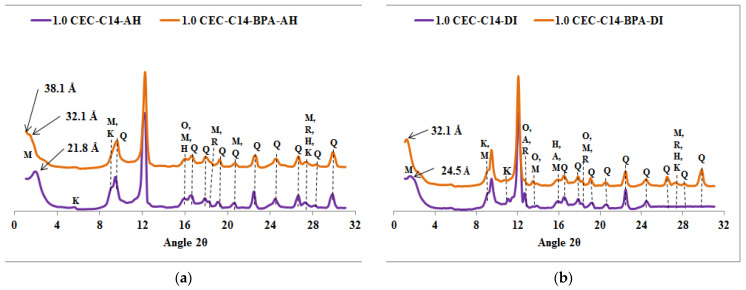
XRPD patterns of organoclays samples before and after BPA adsorption: (**a**) C_14_-AH, (**b**) C_14_-DI. Q = Quartz, M = Montmorillonite, K = Kaolinite, H = Hematite, R = Rutile, O = Orthoclase and A = Anorthite.

**Figure 4 molecules-27-04343-f004:**
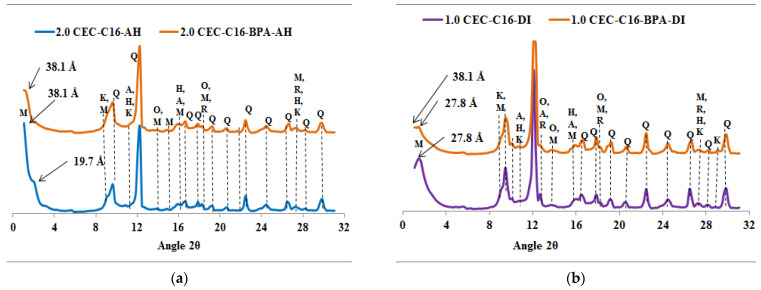
XRPD patterns of organoclays samples before and after BPA adsorption: (**a**) C_16_-AH, (**b**) C_16_-DI. Q = Quartz, M = montmorillonite, K = Kaolinite, H = Hematite, R = Rutile, O = Orthoclase and A = Anorthite.

**Figure 5 molecules-27-04343-f005:**
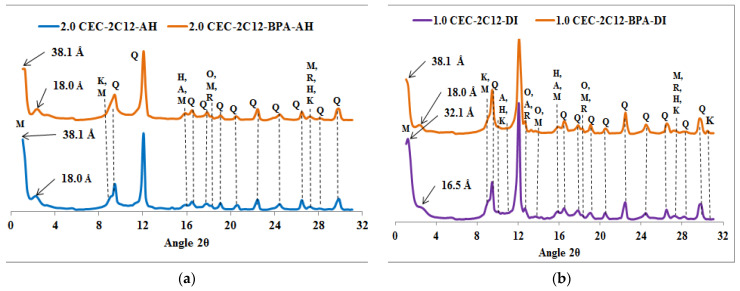
XRPD patterns of organoclays samples before and after BPA adsorption: (**a**) 2C_12_-AH, (**b**) 2C_12_-DI. Q = Quartz, M = montmorillonite, K = Kaolinite, H = Hematite, R = Rutile, O = Orthoclase and A = Anorthite.

**Figure 6 molecules-27-04343-f006:**
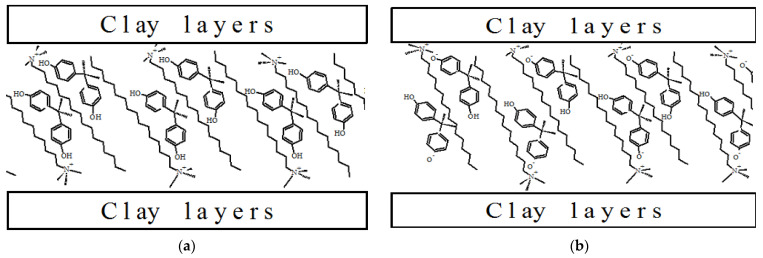
Interaction mechanisms of cationic surfactants with neutral BPA molecules (**a**) and ionized BPA (**b**).

**Figure 7 molecules-27-04343-f007:**
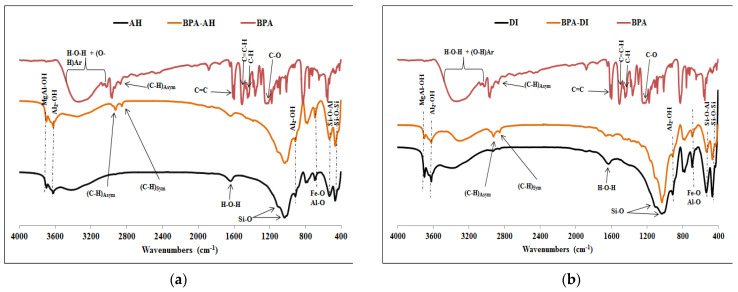
IR spectra of (**a**) AH, BPA, and BPA-AH; (**b**) DI, BPA, and BPA-DI.

**Figure 8 molecules-27-04343-f008:**
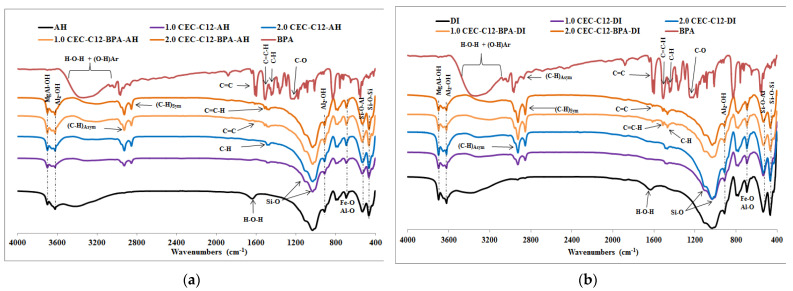
IR spectra of (**a**) AH, BPA, C_12_-AH, and C_12_-BPA-AH; (**b**) DI, BPA, C_12_-DI, and C_12_-BPA-DI.

**Figure 9 molecules-27-04343-f009:**
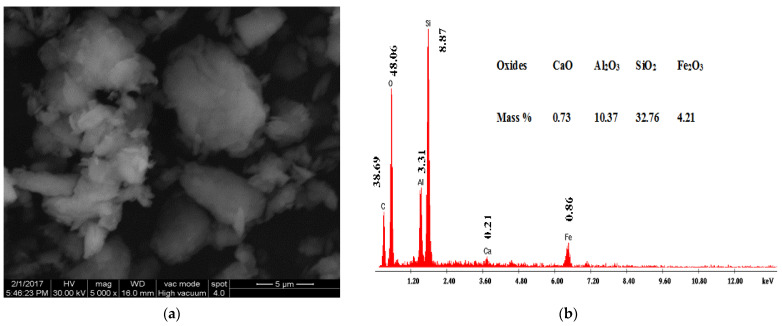
SEM micrographs (**a**) and EDAX spectra (**b**) of AH raw clay.

**Figure 10 molecules-27-04343-f010:**
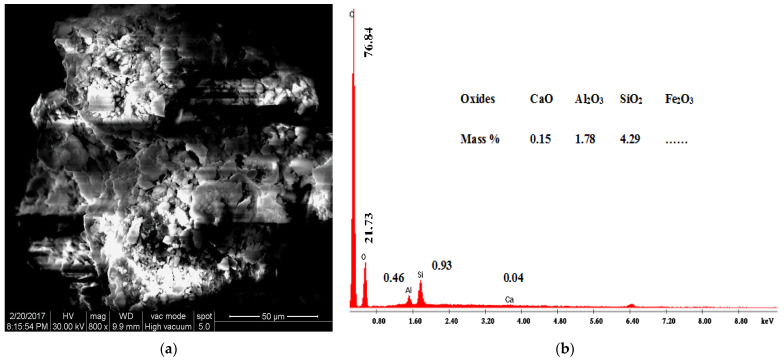
SEM micrographs (**a**) and EDAX spectra (**b**) of 1.0 CEC-C_12_-AH organoclays.

**Figure 11 molecules-27-04343-f011:**
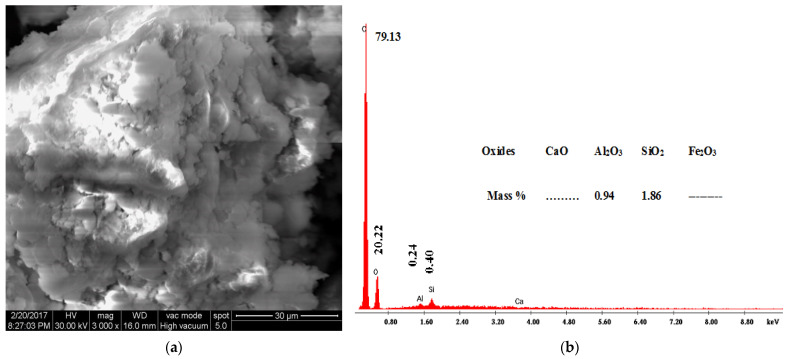
SEM micrographs (**a**) and EDAX spectra (**b**) of 1.0 CEC-C_12_-BPA-AH organoclays.

**Table 1 molecules-27-04343-t001:** Wavenumbers of symmetric and asymmetric stretching vibration of CH_2_ groups of surfactants before and after BPA adsorption.

**Wavenumbers (cm^−1^) of Asymmetric and Symmetric CH_2_ Stretching into the AH Clay**									
**Loading Levels**	**Surfactants**	**C_12_**		**C_14_**		**C_16_**		**2C_12_**	
	**Vibrations**	**CH_2_Asym**	**CH_2_Sym**	**CH_2_Asym**	**CH_2_Sym**	**CH_2_Asym**	**CH_2_Sym**	**CH_2_Asym**	**CH_2_Sym**
1.0-CEC	Before adsorption	2926	2854	2923	2852	2921	2851	2924	2854
2.0-CEC		2925	2854	2921	2851	2920	2850	2921	2853
1.0-CEC	After adsorption	2926	2855	2925	2854	2923	2852	2925	2854
2.0-CEC		2926	2855	2924	2853	2920	2851	2925	2854
**Wavenumbers (cm^−1^) of Asymmetric and Symmetric CH_2_ Stretching into the DI clay**									
**Loading Levels**	**Surfactants**	**C_12_**		**C_14_**		**C_16_**		**2C_12_**	
	**Vibrations**	**CH_2_Asym**	**CH_2_Sym**	**CH_2_Asym**	**CH_2_Sym**	**CH_2_Asym**	**CH_2_Sym**	**CH_2_Asym**	**CH_2_Sym**
1.0-CEC	Before adsorption	2926	2855	2924	2853	2922	2851	2925	2855
2.0-CEC		2924	2854	2922	2852	2920	2851	2924	2854
1.0-CEC	After adsorption	2925	2854	2926	2854	2921	2852	2926	2855
2.0-CEC		2925	2854	2925	2854	2921	2851	2926	2855
Surfactants pure		2918	2850	2918	2849	2919	2849	2921	2853

**Table 2 molecules-27-04343-t002:** Comparison of the present results with those reported in literature.

Types of Adsorbents	XRPD Analyses	Type of BPA Retention Mechanism	Type of Interactions	References
Organo-montmorillonite	Not presented or not done	Inner and outer retention	- Interaction of positively charged head group of the surfactants with the oxygen atoms of a phenol group in neutral BPA; - BPA anions tended to associate with the positively charged head of surfactants on both inner and outer layers	Park et al. [8]
Surfactant-modified zeolite	Not presented or not done	Inner and outer retention	- Interaction of the positively charged head of C_16_ surfactant with the oxygen atoms of a phenol groups; - Interaction of BPA anions with the positively charged ‘‘head” of C_16_ surfactant on both inner and outer layers; - BPA molecules point inside the hydrophobic phase of C_16_ surfactant.	Dong et al. [17]
Amphoteric surfactant activated montmorillonite	Marginal change in the *d*_001_ basal spacing	BPA is not inserted into the interlayers: outer retention	- BPA tended to associate with the hydrophobic tail of surfactants on outer layer of the clays	Liu et al. [23]
C_12_^−^, C_14_^−^, C_16_^−^, 2C_12_^−^AH and C_12_^−^, C_14_^−^, C_16_^−^, 2C_12_^−^DI organoclays	Increase in the *d*_001_ basal spacing	- Large parts of BPA molecules are inserted into the Ca-montmorillonite interlayers; - Weak parts of BPA molecules are retained onto the surface of montmorillonite, kaolinite and other impurities	- Interaction of BPA molecular form or mono-ionized form with the positively charged nitrogen of C_12_, C_14_, C_16_, 2C_12_ surfactants on both the inner layer and the outer layer; - Hydroxyl of BPA in the mono-ionized form pointed inside the alkyl long chain	Present study

## Data Availability

Not applicable.

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
