# Peer review of "Sorption of Bisphenol A as Model for Sorption Ability of Organoclays"

_molecules, 2022, doi:10.3390/molecules27144343_

Round 1

Reviewer 1 Report

Comments to the Author Manuscript molecules-1774909

 The results are very interesting and show good chemistry, the paper is compressively well organized. There are some points which must be edited or clarified by providing additional information or a comment. This manuscript is recommended to be published only after including and addressing the below listed comments with major

1. Key words section should be extended and more concretized.

2. Introduction section has to be elaborated and more compressed.

3. Please use only one version XRPD or XRD!

4. Im strongly recommends moving all similar spectra and graphs to the supplementary file (i.e. Figures 1-5 and 7-11).

5. The quality of the Fig 12-14 should be improved and more readable scale bar havw to be corrected. Same suggestion for EDX spectra. Please make it readable.

6. Next data is missing in manuscript and should be added in revised version:

Time dependent sorption of BPA by studied clays

What about kinetics?

Please propose the mechanism of BPA sorption

Did you evaluate the most appropriate model of BPA sorption?

7. Comparative table with results from a similar previously conducted studies should be added.

8. Conclusion have to be elaborated

What about statistical analysis? In temperature range of 250-400 °C experimental graphs are too closed to each other…And it rather difficult to clamed about significant differences. Authors have to clarify this quote.

4. Did authors evaluate a reusability of prepared catalysts for several runs?

5. Authors have to add comparative data on studied catalysts from related publications.

Our decision on this manuscript – Major revision. After making substantial changes in article it could be recommended for publication

Author Response

Sorption of bisphenol A as model for sorption ability of organoclays

Issaka Garikoé1, Boubié Guel1,* and Ingmar Persson2

Response to Reviewer 1 Comments

The results are very interesting and show good chemistry, the paper is compressively well organized. There are some points which must be edited or clarified by providing additional information or a comment. This manuscript is recommended to be published only after including and addressing the below listed comments with major revisions:

Point 1: Key words section should be extended and more concretized.

Response 1: Keywords have been extended and concretized. The extended keywords are as follows:

Keywords: Smectite; Organoclays; Surfactant; Intercalation, Hydrophobicity; Rearrangement, Expansion; Interaction; Partitioning process; Ca-Montmorillonite

Point 2: Introduction section has to be elaborated and more compressed.

Response 2: The Introduction section has been more compressed and elaborated. We moved some parts to the Supporting Information (SI) section. In the new version the rephrased text is given in red color.

Point 3: Please use only one version XRPD or XRD!

Response 3: We now use one version XRPD in the entire text.

Point 4: Im strongly recommends moving all similar spectra and graphs to the supplementary file (i.e. Figures 1-5 and 7-11).

Response 4: We appreciate the recommendation of the reviewer. The following modifications have been undertaken:

  • 1 Table 1 has been moved into SI section while maintaining figures 1-5 which are not similar in the main text. These figures correspond to the main XRPD diffractogramms of the natural raw clays and the organoclays before and after BPA sorption.
  • 2 Infrared spectra Figures: Figures 7 and 8 are maintained in the main text with some modifications in the legends, while Figures 9-11 have been moved into the SI section:

Figure 7. IR spectra of (a) AH, BPA, BPA-AH, (b) DI, BPA, BPA-DI.

Figure 8. IR spectra of (a) AH, BPA, C12-AH, C12-BPA-AH, (b) DI, BPA, C12-DI, C12-BPA-DI.

Figure SI2. IR spectra of (a) AH, BPA, C14-AH, C14-BPA-AH, (b) DI, BPA, C14-DI, C14-BPA-DI.

Figure SI3. IR spectra of (a) AH, BPA, C16-AH, C16-BPA-AH, (b) DI, BPA, C16-DI, C16-BPA-DI.

Figure SI4. IR spectra of (a) AH, BPA, 2C12-AH, 2C12-BPA-AH, (b) DI, BPA, 2C12-DI, 2C12-BPA-DI.

Point 5: The quality of the Fig 12-14 should be improved and more readable scale bar havw to be corrected. Same suggestion for EDX spectra. Please make it readable.

Response 5: We appreciate the recommendation of the reviewer. The following modifications have been undertaken: only one SEM image is now presented out of Figures 12-14. The scale is now readable which gives improved Figures. The same modifications have also been undertaken for Figures of the EDX spectra.

Point 6: Next data is missing in manuscript and should be added in revised version:

Time dependent sorption of BPA by studied clays

What about kinetics?

Please propose the mechanism of BPA sorption

Did you evaluate the most appropriate model of BPA sorption?

Response 6: The above mentionned data (time dependent sorption of BPA by studied clays, kinetics, mechanism of BPA sorption, the most appropriate model of BPA sorption) have already been the subject of our previous paper published in Bioremediation Journal  (Garikoé, I., B. Sorgho, A. Yaméogo, B. Guel and D. Andala. Removal of bisphenol A by adsorption on organically modified clays from Burkina Faso. Bioremediation Journal. 2021, 25 (1): 22–47. doi:10.1080/10889868.2020.1842321).

In this paper, complete and full experimental investigations on batch adsorption regarding the effects of sorbent dose, agitation time, speed and kinetics, initial concentration and isotherm, pH, surfactant loading levels, temperature, and thermodynamics have been undertaken.

We reported that the sorption mechanism of BPA on raw clays and organoclays is best described by pseudo-second-order kinetics and the Langmuir isotherm model, with maximum sorption capacity in acidic solution. We also found that physisorption and chemisorption might be involved in the BPA sorption and this was explained by the short time observed to reach equilibrium, and the pseudo second order kinetics. It has also been shown that pH of the reaction mixture is the most important factor to determine the sorption properties of the sorbent, and acidic conditions seem ideal for the sorption process. These previous results were shortly addressed in the manuscript as follows:

“Previous studies reported that the sorption mechanism of BPA on organoclays is best described by pseudo-second-order kinetics and the Langmuir isotherm model, with maximum sorption capacity in acidic solution [8,11,13,16,17]. In our previous, study we profoundly investigated the thermodynamic and kinetic parameters and found that physisorption and chemisorption might be involved in the BPA adsorption mechanism [11].”

After the batch adsorption investigations on BPA adsorption on organoclays, it is of general interest to carry out experimental investigations to find how BPA molecules are arranged into the organoclays’interlayers, and how the surfactants structurally rearrange after BPA sorption. As a matter of fact, the present investigations constitute a follow-up of our previous paper [11] and aims to provide detailed experimental results for a better understanding of the interfaces BPA sorbate/cationic surfactants/Ca-montmorillonite. The structural geometry and texture of organoclays were probed using XRPD analysis. The molecular arrangement of the intercalated surfactants and BPA molecules could be interpreted by FT-IR investigations. The morphological changes of the clay composites are assessed by SEM pictures.

The present work is concerned with experimental data (XRPD, FT-IR and SEM) carried out on the spent organoclays to obtain information on the molecular arrangement in the organoclays’interlayers and to examine the rearrangements of the surfactants after the second BPA adsorption. At this step, we have proposed an arrangement which is based on previous models by Dong et al. [17] on BPA sorption onto surfactant modified zeolites and Kamitori et al. [33] on the C12 alkyl chains and p-phenylphenol complex study. Although, we do not have the theoretical data to assess the most appropriate model of BPA sorption, it is possible based on previous models to propose such an arrangement. It can be seen in the proposed arrangement that for the neutral forms of BPA (pH≤8), the hydroxyl groups (-OH) of the phenol ring in the BPA molecules tend to interact with the positively charged nitrogen of the surfactant (Figure 6.a). In the case of the BPA mono-ionized forms (pH>8), the -O- groups of BPA tend to join the positively charged nitrogen of surfactant, and the hydroxyl group (-OH) pointed inside of the hydrophobic phase created by the long alkyl chains (figure 6.b). A comparative table has been added in the manuscript to show what has previously been done and what we propose.

Point 7: Comparative table with results from a similar previously conducted studies should be added.

Response 7: A comparative table has been added in the manuscript to take into account previously studies on similar systems (See new Table 2 in the revised manuscript) in the new paragraph 3.2

3.2. Comparison of the results with previously studies on similar systems

Few studies have proposed arrangements of BPA molecules in the inner/outer layers of organoclays [8,23] or organozeolite [17]. BPA molecules retention mechanism and their interactions in the organoclays’ interlayers are interpreted in different ways in the literature [8,17,23]. Table 2 summarizes the comparison of the results obtained in the present work with previously studies on similar systems. The previous reported investigations indicated that BPA removal from aqueous solution occurred through inner and outer retention mechanism along with a marginal change in the d001 basal spacing. The present work shows a significant increase in the d001 basal spacing in the montmorillonite phase, which suggests that the adsorption of BPA molecules is mainly due to their insertion into the montmorillonite interlayers. In this context, it can be concluded that, in the BPA removal by using these natural clays modified with these surfactants, the inner retention is predominantly favored compared to the outer retention.

Point 8: Conclusion have to be elaborated

Response 8: The conclusion has been elaborated as described in the text in red color.

Last point of the reviewer comments:

What about statistical analysis? In temperature range of 250-400 °C experimental graphs are too closed to each other…And it rather difficult to claim about significant differences. Authors have to clarify this quote.

  1. Did authors evaluate a reusability of prepared catalysts for several runs?
  2. Authors have to add comparative data on studied catalysts from related publications.

Response: We are not concerned with the points related to temperature effect since our work did not investigate the effect of temperature in the range of 250-400 °C.

To fully assess the practical implications of the results described in this paper, the fate of the adsorbed BPA contaminant need to be further investigated. The objective will be to assess the possible re-use of the spent organoclays. The reusability of the spent organoclays is part of our next paper which will be soon submitted.

Reviewer 2 Report

1. The current manuscript is not well lacks novelity   and already there are works which have been published on similar topic like

“Adsorption of bisphenol A from aqueous solution onto organoclay: Experimental design, kinetic, equilibrium and thermodynamic study”,Powder Technology, 2022

“Bisphenol A sorption by organo-montmorillonite: Implications for the removal of organic contaminants from water”, Chemosphere, 2014

2. The title does not supports the work done in the manuscript. The  point the authors highlight in this manuscript is “XRPD, FT-IR, and SEM analyses provide in-depth information regarding BPA arrangements and interactions with surfactants in the organoclays interlayers. Whereas the basic adsorption experiments are not included in this work. The authors change the title in such a way that it highlights what the authors actually want to talk about.

3. The authors have tried to use three characterization techniques for explaining the interacts of the BPA, surfactants and the organoclay which is not enough for a high impact journal like Molecules. The authors should include other techniques like BET-BJH analysis to show the effect of porosity, TGA studies etc.

4. The SEM images are not clear and small to actually understand the surface morphology.

5. A figure showcasing the mechanism should be proposed based on the characterization

Author Response

Sorption of bisphenol A as model for sorption ability of organoclays

Issaka Garikoé1, Boubié Guel1,* and Ingmar Persson2

Response to Reviewer 2 Comments

Point 1: The current manuscript is not well lacks novelity and already there are works which have been published on similar topic like

“Adsorption of bisphenol A from aqueous solution onto organoclay: Experimental design, kinetic, equilibrium and thermodynamic study”, Powder Technology, 2022

“Bisphenol A sorption by organo-montmorillonite: Implications for the removal of organic contaminants from water”, Chemosphere, 2014

Response 1: We greatly appreciate the reviewer comments. As a matter of fact, this work is part of a global research on the adsorption of bisphenol A by surfactants modified natural smectites clays minerals in order to purify waste water. Significant results have already been obtained by our research group as for “Adsorption of bisphenol A from aqueous solution onto organoclay: Experimental design, kinetic, equilibrium and thermodynamic study” and the results have been published in abstracts and indexed Journals. This paper constitutes indeed the follow-up and continuation of our previous papers. In the following, we’ll give an overview of what we have already done and what we envisage to do. This global work is divided in four steps:

The first step of our work was concerned with the solid-state synthesis of organoclays by using two natural smectites clays and four alkylammonium surfactants n-dodecyl­trimethylammonium bromide, (n-C12H25(CH3)3NBr), n-tetradecyltrimethylammonium bro­mide, (n-C14H29(CH3)3NBr), n-hexadecyltrimethylammonium bromide (n-C16H33(CH3)3NBr) and di-n-dodecyldimethylammonium bromide ((n-C12H25)2(CH3)2NBr) at different levels of the cation exchange capacity (CEC). The synthesized organo-smectites were characterized regarding relative density, structural and textural properties. The results have been the subject of the following published paper:

  1. Garikoé, I., B. Sorgho, B. Guel and I. Persson. Solid-state synthesis and physico-chemical characterization of modified smectites using natural clays from Burkina Faso. Bull. Chem. Soc. Ethiop. 2021, 35 (1): 43-59. doi:10.4314/bcse.v35i1.4

Significant results were obtained in this paper. Measurements showed that the relative density of the organoclays decreases with increasing surfactant loading. Thus, the natural clays swell substantially at treatment with surfactants. The increase of the d001 spacing shown by XRPD proved the swelling of layers by intercalation of the surfactants. The value of basal spacing reaches 25 Å for modified clays treated with C12, 32 Å for those treated with C14, and more than 38 Å for those treated with C16 and 2C12 at 2.0 CEC loading level of the surfactant. The IR spectra showed the CH2 asymmetric and symmetric stretching bands at around 2920 and 2850 cm-1, respectively, proved successful intercalation. FTIR spectra showed that the wavenumbers of asymmetric and symmetric stretching vibrations of the surfactants in organoclays decrease and approach the values observed for the pure surfactants with increasing surfactant loading. The intercalation of the quaternary alkylammonium cations into the clays interlayers causes that the hydrophilic properties of raw clays change to hydrophobic and lipophilic. The solid-state intercalation of alkyltrimethylammonium and di-alkyldimethylammonium cations is efficient. Although numerous studies have shown that the behavior and properties of organoclays strongly depend on their structure and the environment of the organic molecules between the clay sheets, few studies have undertaken relative density measurements. Our investigations showed that the relative density of the organoclays decreased with increased length and number of the alkyl chains and with increased surfactant loading. Compared to existing literature, this was an extended investigation including simultaneously three alkyl-trimethylammonium and one di-alkyl-dimethylammonium surfactants and two natural clays. The innovations in the study was the use of organoclays prepared via solid-state methodology. Indeed, few studies have been devoted to such organoclays prepared via solid-state methodology.

The second step of our work was concerned with the removal of BPA by using the surfactants modified organoclays, synthesized by solid-state synthesis. In this paper, complete and full experimental investigations on batch adsorption regarding the effects of sorbent dose, agitation time, kinetics, initial concentration and isotherms, pH, surfactant loading levels, temperature, and thermodynamics have been undertaken.

We reported that the sorption mechanism of BPA on raw clays and organoclays is best described by pseudo-second-order kinetics and the Langmuir isotherm model, with maximum sorption capacity in acidic solution. We also found for the first time that physisorption and chemisorption might be involved in the BPA sorption and this was explained by the short time required to reach equilibrium and the pseudo second order kinetics. It has also been shown that pH of the reaction mixture is the most important factor to determine the sorption properties of the sorbent, and acidic conditions seem ideal for the sorption process. To the best of our knowledge, work on the efficiency of organically modified clays, synthesized via solid-state intercalation of surfactants, on BPA removal from water was scarcely reported in the literature.

The above mentionned data (time dependent sorption of BPA by studied organoclays, kinetics, mechanism of BPA sorption, the most appropriate model of BPA sorption) have already been the subject of our previous paper published in Bioremediation Journal:

  1. Garikoé, I., B. Sorgho, A. Yaméogo, B. Guel and D. Andala. Removal of bisphenol A by adsorption on organically modified clays from Burkina Faso. Bioremediation Journal. 2021, 25 (1): 22–47. doi:10.1080/10889868.2020.1842321.

The third step of our work constitutes the results of the present work which is concerned with the experimental data (XRPD, FT-IR and SEM) to obtain information on the molecular arrangement in the organoclay’s interlayers and to examine the rearrangements of the surfactants after the second BPA adsorption. As a matter of fact, this is the reason why the title “Sorption of bisphenol A as model for sorption ability of organoclays” has been chosen since it is appropriate to the results that were obtained.

At this step, we have proposed an arrangement which is based on previous models by Dong et al. [17] on BPA sorption onto surfactant modified zeolites and Kamitori et al. [33] on the C12 alkyl chains and p-phenylphenol complex study. Although, we do not have the theoretical data, it is possible based on previous models to propose such an arrangement. It can be seen in the proposed arrangement that for the neutral forms of BPA (pH≤8), the hydroxyl groups (-OH) of the phenol ring in the BPA molecules tend to interact with the positively charged nitrogens of the surfactant (Figure 6.a). In the case of the BPA mono-ionized forms (pH>8), the -O- groups of BPA tend to join the positively charged nitrogen of surfactant, and the hydroxyl group (-OH) pointed inside of the hydrophobic phase created by the long alkyl chains (Figure 6.b).

The fourth step of our work will envisage the theoretical data on the geometry of the arrangement of ions in the interlayer space, which is influenced by the layer charge and the length of alkyl chains, and also data on what factors can influence the change in the basal space. These data are currently under investigations and will be the subject of our forthcoming paper.

Point 2: The title does not supports the work done in the manuscript. The point the authors highlight in this manuscript is “XRPD, FT-IR, and SEM analyses provide in-depth information regarding BPA arrangements and interactions with surfactants in the organoclays interlayers. Whereas the basic adsorption experiments are not included in this work. The authors change the title in such a way that it highlights what the authors actually want to talk about.

Response 2: As we have already mentioned, the present paper is a follow-up and continuation of our previous paper which dealt with the basic adsorption experiments such as the effects of sorbent dose, agitation time, kinetics, initial concentration and isotherms, pH, surfactant loading, temperature, and thermodynamics. In the present paper, we refer to the results of this previous paper (see reference 11). As a matter of fact, the title of this published paper was “Removal of bisphenol A by adsorption on organically modified clays from Burkina Faso”. After the batch adsorption investigations on BPA adsorption on organoclays, it is generally of great interest to carry out experimental investigations to find how BPA molecules are arranged into the organoclays’interlayers and how the surfactants rearrange after BPA sorption. This is the topic of the present paper and this why the paper is entitled " Sorption of bisphenol A as model for sorption ability of organoclays”.

Point 3: The authors have tried to use three characterization techniques for explaining the interacts of the BPA, surfactants and the organoclay which is not enough for a high impact journal like Molecules. The authors should include other techniques like BET-BJH analysis to show the effect of porosity, TGA studies etc.

Response 3: We greatly appreciate the reviewer comments. It is worth to highlight the fact that XRPD analysis, infrared spectroscopy and SEM analysis are among the most informative methods to investigate the adsorption of BPA on organoclays and to obtain information on the molecular arrangement in their interlayers and finally to examine the rearrangements of the surfactants after the second BPA adsorption. Indeed, in the literature the structural geometry and texture of organoclays are probed using XRPD analysis. The molecular arrangement of the intercalated surfactants and BPA molecules are carefully investigated by FT-IR investigations. Finally, the morphological changes of the clay composites are assessed by SEM pictures. Although other techniques like BET-BJH analysis and TGA studies techniques are utilized for the characterization of organoclays, they are less used to probe the molecular arrangement of surfactants intercalated into organoclays’s interlayers, and to probe the BPA molecules interactions with the intercalated surfactants. 

Point 4: The SEM images are not clear and small to actually understand the surface morphology.

Response 4: We appreciate the recommendation of the reviewer. The following modifications have been undertaken: only one SEM image is now presented out of Figures 12-14. The scale is now readable which leads to improved Figures. The same modifications have also been undertaken for the figures of the EDX spectra.

Point 5: A figure showcasing the mechanism should be proposed based on the characterization

Response 5: Based on the characterizations, several mechanisms have been proposed.

In the first paper, depending on the long alkyl chains the mechanism of orientation of the quaternary ammonium ions surfactant has been proposed as follows: “The orientation of the intercalated quaternary ammonium ions changes from being parallel with the sheets to become more and more upright with increasing loading”, see the following Figure. This mechanism can be found in our first paper “25. Garikoé, I., B. Sorgho, B. Guel and I. Persson. Solid-state synthesis and physico-chemical characterization of modified smectites using natural clays from Burkina Faso. Bull. Chem. Soc. Ethiop. 2021, 35 (1): 43-59. doi:10.4314/bcse.v35i1.4”

Figure 3. Surfactant molecules orientations into the interlayer space: arrangements (a) monolayer, (b) bilayers, and (c, d) pseudo-trilayers and (e, f and g, h) paraffin-type for one and two long alkyl chains respectively.

In the second paper, kinetic and thermodynamic data showed that in the mechanism of BPA removal chemisorption and physorption may be involved.

In the present paper, Figure 6 in the manuscript describes the possible arrangement of BPA molecules into the orgaanoclays’ interlayers. The proposed arrangement is based on previous models by Dong et al. [17] on BPA sorption onto surfactant modified zeolites and Kamitori et al. [33] on the C12 alkyl chains and p-phenylphenol complex study. Although, we do not have the theoretical data, it is possible based on previous models to propose such an arrangement. It can be seen in the proposed arrangement that for the neutral forms of BPA (pH≤8), the hydroxyl groups (-OH) of the phenol ring in the BPA molecules tend to interact with the positively charged nitrogen of the surfactant (Figure 6.a of the present manuscript). In the case of the BPA mono-ionized forms (pH>8), the -O- groups of BPA tend to join the positively charged nitrogen of surfactant, and the hydroxyl group (-OH) pointed inside of the hydrophobic phase created by the long alkyl chains (Figure 6.b of the present manuscript). The new figure 6 has been improved as shown in the manuscript. A comparative table has been added in the manuscript to show what has previously been done and what we propose.

(a)

(b)

Figure 6. Interaction mechanisms of cationic surfactants with neutral BPA molecules (a) and ionized BPA (b).

3.2. Comparison of the results with previously studies on similar systems

Few studies have proposed arrangements of BPA molecules in the inner/outer layers of organoclays [8,23] or organozeolite [17]. BPA molecules retention mechanism and their interactions in the organoclays’ interlayers are interpreted in different ways in the literature [8,17,23]. Table 2 summarizes the comparison of the results obtained in the present work with previously studies on similar systems. The previous reported investigations indicated that BPA removal from aqueous solution occurred through inner and outer retention mechanism along with a marginal change in the d001 basal spacing. The present work shows a significant increase in the d001 basal spacing in the montmorillonite phase, which suggests that the adsorption of BPA molecules is mainly due to their insertion into the montmorillonite interlayers. In this context, it can be concluded that, in the BPA removal by using these natural clays modified with these surfactants, the inner retention is predominantly favored compared to the outer retention.

Reviewer 3 Report

The authors present an interesting manuscript that shows changes in the structure of clays and organoclays during the sorption of bisphenol studied by three different methods. Despite the fact that some conclusions of this manuscript are controversial, it contains important experimental data and may be published in the Journal after revision.

Notes on the text of the manuscript:

1) It is a pity that the authors worked with natural clays, which contain numerous impurities. Some of them, such as kaolinite and hematite, can also be involved in adsorption.

2) In the Abstracts – “bisphenol-A”, in the title and text – “bisphenol A”.

3) In section 2.2. it is necessary to give a detailed procedure for obtaining organoclays and adsorption experiments. Linking to the previous paper is not enough. Reviewers may have questions when they get acquainted with the actually missing methods.

4) Table 2. Why are there two values ​​in some positions?

5) Line 238-242. What pH are we talking about - equilibrium or initial solutions? An aqueous solution of bentonite clays usually has an alkaline pH.

6) Figures 1-5 actually duplicate the data of Table 1.

7) Figure 6. The geometry of the arrangement of ions in the interlayer space is influenced by the layer charge and the length of alkyl chains. Therefore, the indicated arrangement of various molecules in the interlayer space is speculative; it has not been proven in any way. In addition, the presence of a large number of molecules in the figure makes it difficult to understand.

8) It is necessary to provide theoretical data on what factors can influence the change in the basal space.

Author Response

Sorption of bisphenol A as model for sorption ability of organoclays

Issaka Garikoé1, Boubié Guel1,* and Ingmar Persson2

Response to Reviewer 3 Comments

The authors present an interesting manuscript that shows changes in the structure of clays and organoclays during the sorption of bisphenol studied by three different methods. Despite the fact that some conclusions of this manuscript are controversial, it contains important experimental data and may be published in the Journal after revision.

Notes on the text of the manuscript:

Point 1: It is a pity that the authors worked with natural clays, which contain numerous impurities. Some of them, such as kaolinite and hematite, can also be involved in adsorption.

Response 1: We appreciate the reviewer comments. It is indeed true that natural clays contain numerous impurities such as kaoloinite and hematite that are likely to be involved in the BPA sorption. As a matter of fact, according to the literature, mineral phases like kaolinite and hematite can intervene in the adsorption process. However, this adsorption remains lower compared to the main and dominant mineral phase which is the Ca-montmorillonite, which allows high removal levels of BPA. The results showed that efficient adsorption could be obtained with these natural clays modified with these surfactants. Similar results were also described in the literature regarding the efficiency of surfactants modified natural clays. Moreover, this work constitutes a first set of investigations to establish the potential of these modified natural clays to remove an organic pollutant such as BPA. The next step of our work will be to purify theses natural clays in order to remove several impurities which are likely to be involved in the adsorption process. The following text has been added in the revised manuscript (See X-ray diffraction Section).

“It is worth to mention the fact that in addition to montmorillonite, AH and DI clays contain non-swelling phases such as kaolinite, orthoclase, anorthite and other minerals phases such as quartz, rutile, and hematite. However, literature results indicated that all these phases contributed weakly to the removal of organic pollutants. For example, Styszko et al., Alkaram et al. reported that unmodified montmorillonite, bentonite or kaolinite had a low adsorption capacity toward phenolic compounds [28,29]. According to the work by Li et al., iron oxides such as hematite (α-Fe2O3), maghemite (γ-Fe2O3), lepidocrocite (γ-FeOOH) presented a low adsorption capacity toward BPA (2.7%) in absence of a UV irradiation [30]. However, UV irradiation caused a photo-degradation of the BPA molecules. Kaplan et al. reported a weak percentage of BPA adsorption (5%) by rutile from aqueous solution of 10 mg/L of BPA as initial concentration [31]. The authors brought back an improvement of the rate of mineralization of BPA (51%) in the presence of UV irradiation. Taking into consideration all these results, it can be concluded that hematite and rutile which are present in our AH and DI clays would contribute weakly to the adsorption of BPA. As a result, the removal of BPA is mainly due to its interaction with the surfactants intercalated into the organoclays’interlayers, as already reported [11].”

Point 2: In the Abstracts – “bisphenol-A”, in the title and text – “bisphenol A”.

Response 2: “bisphenol-A” is replaced by “bisphenol A” in the entire manuscript text.

Point 3: In section 2.2. it is necessary to give a detailed procedure for obtaining organoclays and adsorption experiments. Linking to the previous paper is not enough. Reviewers may have questions when they get acquainted with the actually missing methods.

Response 3: We appreciate the reviewer comments. Detailed procedures for obtaining organoclays and adsorption experiments are provided in the SI section.

Point 4: Table 2. Why are there two values ​​in some positions?

Response 4: From the literature, the reflections observed on some of the XRPD diffractogramms are due the fact that there are two basal spacing values as a result of the co-existence of more than one arrangement of the organic molecules inserted into the interlayers. 

Point 5: Line 238-242. What pH are we talking about - equilibrium or initial solutions? An aqueous solution of bentonite clays usually has an alkaline pH.

Response 5: We refer to the pH of BPA initial solution.

Point 6: Figures 1-5 actually duplicate the data of Table 1.

Response 6: We appreciate the recommendation of the reviewer. The following modifications have been undertaken:

  • Table 1 has been moved into SI section while maintaining the figures 1-5 which are not similar in the main text. These figures correspond to the main XRPD diffractogramms of the natural raw clays and the organoclays before and after BPA sorption.
  • Infrared spectra Figures: Figure 7 and 8 are maintained in the main text with some modifications in the legends, while the other Figures 9-11 have been moved into SI section:

Figure 7. IR spectra of (a) AH, BPA, BPA-AH, (b) DI, BPA, BPA-DI.

Figure 8. IR spectra of (a) AH, BPA, C12-AH, C12-BPA-AH, (b) DI, BPA, C12-DI, C12-BPA-DI.

Figure SI2. IR spectra of (a) AH, BPA, C14-AH, C14-BPA-AH, (b) DI, BPA, C14-DI, C14-BPA-DI.

Figure SI3. IR spectra of (a) AH, BPA, C16-AH, C16-BPA-AH, (b) DI, BPA, C16-DI, C16-BPA-DI.

Figure SI4. IR spectra of (a) AH, BPA, 2C12-AH, 2C12-BPA-AH, (b) DI, BPA, 2C12-DI, 2C12-BPA-DI.

Point 7: Figure 6. The geometry of the arrangement of ions in the interlayer space is influenced by the layer charge and the length of alkyl chains. Therefore, the indicated arrangement of various molecules in the interlayer space is speculative; it has not been proven in any way. In addition, the presence of a large number of molecules in the figure makes it difficult to understand.

Point 8: It is necessary to provide theoretical data on what factors can influence the change in the basal space.

Response 7 and Response 8: We greatly appreciate the reviewer comments. As a matter of fact, this work is part of a global research on the adsorption of bisphenol A by surfactants modified natural smectites clays minerals in order to purify waste water. Significant results have already been obtained as for “Adsorption of bisphenol A from aqueous solution onto organoclay: Experimental design, kinetic, equilibrium and thermodynamic study” and the results have been published in abstracts and indexed Journals. This paper constitutes indeed the follow-up and continuation of our previous papers. In the following, we’ll give an overview of what we have already done and what we envisage to do. The global work is divided in four steps:

The first step of our work was concerned with the solid-state synthesis of organoclays by using two natural smectites clays and four alkylammonium surfactants n-dodecyl­trimethylammonium bromide, (n-C12H25(CH3)3NBr), n-tetradecyltrimethylammonium bro­mide, (n-C14H29(CH3)3NBr), n-hexadecyltrimethylammonium bromide (n-C16H33(CH3)3NBr) and di-n-dodecyldimethylammonium bromide ((n-C12H25)2(CH3)2NBr) at different levels of the cation exchange capacity (CEC). The synthesized organo-smectites were characterized regarding relative density, structural and textural properties. The results have been the subject of the following published paper:

  1. Garikoé, I., B. Sorgho, B. Guel and I. Persson. Solid-state synthesis and physico-chemical characterization of modified smectites using natural clays from Burkina Faso. Bull. Chem. Soc. Ethiop. 2021, 35 (1): 43-59. doi:10.4314/bcse.v35i1.4

The second step was concerned with the removal of BPA by using the organoclays modified with surfactants, synthesized by solid-state synthesis. In this paper, complete and full experimental investigations on batch adsorption regarding the effects of sorbent dose, agitation time, speed and kinetics, initial concentration and isotherm, pH, surfactant loading levels, temperature, and thermodynamics have been undertaken.

We reported that the sorption mechanism of BPA on raw clays and organoclays is best described by pseudo-second-order kinetics and the Langmuir isotherm model, with maximum sorption capacity in acidic solution. We also found that physisorption and chemisorption might be involved in the BPA sorption and this was explained by the short time required to reach equilibrium and the pseudo second order kinetics. It has also been shown that pH of the reaction mixture is the most important factor to determine the sorption properties of the sorbent, and acidic conditions seem ideal for the sorption process.

The above mentioned data (time dependent sorption of BPA by studied organoclays, kinetics, mechanism of BPA sorption, the most appropriate model of BPA sorption) have already been the subject of our previous paper published in Bioremediation Journal:

  1. Garikoé, I., B. Sorgho, A. Yaméogo, B. Guel and D. Andala. Removal of bisphenol A by adsorption on organically modified clays from Burkina Faso. Bioremediation Journal. 2021, 25 (1): 22–47. doi:10.1080/10889868.2020.1842321.

The third step constitutes the results of the prsent work which is concerned with the experimental data (XRPD, FT-IR and SEM) to obtain information on the molecular arrangement in the organoclays’interlayers and to examine the rearrangements of the surfactants after the second BPA adsorption. At this step, we have proposed an arrangement which is based on previous models by Dong et al. [17] on BPA sorption onto surfactant modified zeolites and Kamitori et al. [33] on the C12 alkyl chains and p-phenylphenol complex study. Although, we do not have the theoretical data, it is possible based on previous models to propose such an arrangement. It can be seen in the proposed arrangement that for the neutral forms of BPA (pH≤8), the hydroxyl groups (-OH) of the phenol ring in the BPA molecules tend to interact with the positively charged nitrogen of the surfactant (Figure 6.a). In the case of the BPA mono-ionized forms (pH>8), the -O- groups of BPA tend to join the positively charged nitrogen of surfactant, and the hydroxyl group (-OH) pointed inside of the hydrophobic phase created by the long alkyl chains (figure 6.b). A comparative table has been added in the manuscript to show what has previously been done and what we propose.

3.2. Comparison of the results with previously studies on similar systems

“Few studies have proposed arrangements of BPA molecules in the inner/outer layers of organoclays [8,23] or organozeolite [17]. BPA molecules retention mechanism and their interactions in the organoclays’ interlayers are interpreted in different ways in the literature [8,17,23]. Table 2 summarizes the comparison of the results obtained in the present work with previously studies on similar systems. The previous reported investigations indicated that BPA removal from aqueous solution occurred through inner and outer retention mechanism along with a marginal change in the d001 basal spacing. The present work shows a significant increase in the d001 basal spacing in the montmorillonite phase, which suggests that the adsorption of BPA molecules is mainly due to their insertion into the montmorillonite interlayers. In this context, it can be concluded that, in the BPA removal by using these natural clays modified with these surfactants, the inner retention is predominantly favored compared to the outer retention.”

The fourth step will envisage the theoretical data on the geometry of the arrangement of ions in the interlayer space, which is influenced by the layer charge and the length of alkyl chains, and also data on what factors can influence the change in the basal space. These data are currently under investigations and will be the subject of our forthcoming paper.

Round 2

Reviewer 1 Report

In revised version of manuscript authors correctly responsed for all  my  queries. Thus in present form this manuscript could be recommended for publication

Reviewer 2 Report

The manuscript can be accepted in the present format